Anatomical indicators of Eucalyptus spp. resistance to Glycaspis brimblecombei (Hemiptera: Aphalaridae)

Del Piero Fernando Henrique Moreno de Oliveira 1
http://orcid.org/0000-0001-9875-4158 Wilcken Carlos Frederico 1
Domingues Maurício Magalhães 1 mauricio.m.domingues@unesp.br
Favoreto Ana Laura 1
Rodella Roberto Antonio 2
Pereira Alexandre Igor Azevedo 3
Silva Wiane Meloni 4
http://orcid.org/0000-0002-0477-4252 Serrão José Eduardo 5
Zanuncio José Cola 6
1 Departamento de Proteção Vegetal, Faculdade de Ciências Agronômicas, Universidade Estadual Paulista (UNESP), Universidade Estadual Paulista , Botucatu, São Paulo , Brazil
2 Instituto de Biociências, Universidade Estadual Paulista (UNESP), Universidade Estadual Paulista , Botucatu, São Paulo , Brazil
3 IF Goiano, Campus Urutaí, Rodovia Geraldo Silva Nascimento, Fazenda Palmital, Instituto Federal Goiano , Urutaí, Goiás , Brazil
4 Departamento de Engenharia Florestal, Universidade Federal de Viçosa, Universidade Federal de Viçosa , Viçosa, Minas Gerais , Brazil
5 Departamento de Biologia Geral, Universidade Federal de Viçosa, Universidade Federal de Viçosa , Viçosa, Minas Gerais , Brazil
6 Departamento de Entomologia/BIOAGRO, Universidade Federal de Viçosa, Universidade Federal de Viçosa , Viçosa, Minas Gerais , Brazil
Gillespie Joseph
Electronic publication date: 2022 May 13
Publication date: 2022
Volume: 10
Electronic Location ID: e13346
Received 2021 Dec 9; Accepted 2022 Apr 6
Copyright: © 2022 Del Piero et al.
Copyright year: 2022
Copyright holder: Del Piero et al.
License: This is an open access article distributed under the terms of the Creative Commons Attribution License, which permits unrestricted use, distribution, reproduction and adaptation in any medium and for any purpose provided that it is properly attributed. For attribution, the original author(s), title, publication source (PeerJ) and either DOI or URL of the article must be cited.
License URL: https://creativecommons.org/licenses/by/4.0/

Keywords: Anatomical leaf characterization, Red gum lerp psyllid, Leaf damage

Funding: Coordenação de Aperfeiçoamento de Pessoal de Nível Superior-Brasil 001 Fundação de Amparo à Pesquisa do Estado de Minas Gerais Programa Cooperativo sobre Proteção Florestal/PROTEF do Instituto de Pesquisas e Estudos Florestais/IPEF The study was financially supported by the following Brazilian agencies “Coordenação de Aperfeiçoamento de Pessoal de Nível Superior-Brasil (CAPES-Finance Code 001)”, “Fundação de Amparo à Pesquisa do Estado de Minas Gerais (FAPEMIG)” and “Programa Cooperativo sobre Proteção Florestal/PROTEF do Instituto de Pesquisas e Estudos Florestais/IPEF”. The funders had no role in study design, data collection and analysis, decision to publish, or preparation of the manuscript.

==============================
The total area of forest crops in Brazil is 9.55 million hectares, of which 7.5 million hectares are Eucalyptus. These crops are the most productive in the world, but may suffer losses due to exotic pests, including Glycaspis brimblecombei Moore (Hemiptera: Aphalaridae) found in Brazil since 2003. Interactions between Eucalyptus plants and insect pests may led to the selection of resistant genotypes. Eucalyptus species are either susceptible or resistant to this pest group, but the damage they suffer needs to be evaluated. The objective was to determine possible leaf anatomy indicators of different Eucalyptus species associated with G. brimblecombei infestations, focusing on plant resistance to this pest. The study was carried out with Eucalyptus camaldulensis, Eucalyptus grandis, Eucalyptus saligna and Eucalyptus urophylla saplings infested or not by G. brimblecombei eggs and nymphs. Eighteen anatomical characteristics of the leaves of these plants were analyzed. The number of stomata on the adaxial and abaxial sides and the glandular area in the central leaf vein are associated with greater or lesser infestation by G. brimblecombei in the Eucalyptus genotypes.

Introduction

Globally, forest crops cover around 294 million hectares (Food & Agriculture Organization of the United Nations, 2020). Brazil accounts for 9.55 million hectares of this area, with 7.5 million being Eucalyptus. Forest plantations in Brazil are among the most productive in the world with 36.8 m³/ha year and with economic, social and environmental importance (Indústria Brasileira de Árvores, 2021). Native and exotic pests can compromise this productivity (Floris et al., 2020; Pereira et al., 2022). Eucalyptus plantations are established in large contiguous areas that provide a significant quantity of food and shelter for insect pests (Wingfield et al., 2008).

Exotic pests, introduced in the last two decades, are causing losses to the Brazilian forestry sector (Paine, Steinbauer & Lawson, 2011; Almeida et al., 2018). In 2003, Glycaspis brimblecombei Moore (Hemiptera: Aphalaridae) was reported in Brazil (Wilcken et al., 2015) and has reduced crop yields (Saliba et al., 2019). This insect feeds only on Eucalyptus species (Wilcken et al., 2015) and leaf rolling and deformation, “witch broom”, dieback and sooty mold are the main features of its infestation (Dittrich-Schroder et al., 2021).

Control methods for G. brimblecombei should focus on breeding and planting resistant eucalypt varieties, especially in areas with large G. brimblecombei populations (Jere et al., 2020). Different environmental conditions influence host plant susceptibility and infestation levels in the field (Ferreira-Filho et al., 2017; Bush, Slippers & Hurley, 2020).

Leaves, allelochemicals (tannins, phenols and waxes), glands that produce essential oils, often rich in terpenoids, hardness (sclorophilia), heterophilia (differentiation between young and mature leaves) and high regrowth of Eucalyptus plants can affect insect damage to this plant, with potential to select for resistant genotypes (Ohmart & Edwards, 1991).

Leaf anatomy is poorly studied and may allow us to understand pest infestations and the development of new tools for their management. Developing integrated psyllid management in Eucalyptus plantations depends on knowledge of plant/insect interactions. The objective of this study was to determine possible indicators based on leaf anatomy of four Eucalyptus species associated with G. brimblecombei infestations. These indicators may be useful in breeding programs for plant resistance to this pest.

Materials and Methods

The study was carried out at the Universidade Estadual Paulista (FCA/UNESP) in Botucatu, São Paulo state, Brazil. Eucalyptus camaldulensis, E. grandis, E. saligna and E. urophylla were planted in 1.5 L pots with an autoclaved mixture of soil: sand: manure (2: 1: 1) and kept in a greenhouse for infestation with G. brimblecombei.

The Eucalyptus species were previously classified according to their response to G. brimblecombei with E. saligna and E. urophylla being resistant, E. grandis tolerant and E. camaldulensis susceptible to damage (Brennan et al., 2001; Pereira et al., 2013; Ribeiro et al., 2015).

Infestation of the Glycaspis brimblecombei on Eucalyptus plants

Glycaspis brimblecombei eggs and nymphs, collected in the field on Eucalyptus leaves, were placed on 25 cm high saplings of this plant in the laboratory. Each of the plants was infested with approximately 40 nymphs and two egg masses (more than 25 eggs each), weekly, for 4 weeks.

Twenty seedlings of each Eucalyptus species were used per treatment, with 10 plants (replications) infested with G. brimblecombei and another 10, as a control, free from the pest. All the plants in the control were sprayed with systemic insecticide (acephate) and the others only with water, to evaluate the effects of mechanical action of the water.

Anatomical characterization of Eucalyptus leaves

Eucalyptus camaldulensis, E. grandis, E. saligna and E. urophylla leaves, infested or not, were analyzed. The samples were one to two leaves from the middle third of each eucalypt sapling, cut in three parts with the middle third analyzed. These samples were placed in formaldehyde + glacial acetic acid + 50% alcohol fixative solution (FAA-50) for 48 h and stored in 70% ethanol (Johansen, 1940). The samples were submerged into glyco-methacrylate resin (Gerrits, 1991) and cut, transversely, in a manual microtome, in the internervural region and in the central rib, with 15 to 25 μm thickness. The pieces were cleared, stained with acid fuchsin (Brennan, Weinbaum & Pinney, 2001) and toluidine blue pH 4.7 and mounted in synthetic resin (O’Brien, Feder & Mccully, 1964).

The thickness and the area with the epidermal, parenchymal and vascular leaf tissues were obtained with the computer program Image Tool 3.0 (UTHSCA) to evaluate the damage by G. brimblecombei on infested leaves. The quantitative anatomy was performed for three plants (replications) per species of Eucalyptus infested or not by G. brimblecombei. Eighteen variables for anatomical characterization of the leaf were evaluated.

Quantitative variables of leaf anatomical characteristics

The 18 variables related to leaf anatomy were: percentages of upper (%UE) and lower (%LE) epidermis, collenchyma (%Col), phloem (%Ph), xylem (%Xy), chlorenchyma (%Chl), gland (%Gl), and total cross-sectional area (mm2) (CS) in the region of the central rib, thickness of the upper (TUE) and lower (TLE) epidermis, upper (TUPP) and lower (TLPP) palisade parenchyma, spongy parenchyma (TSP), leaf (TL), mesophyll (TM), the mean area of a gland (MGA), and number of stomata/mm2 of the upper (NUS) and lower (NLS) surfaces in the internervure region (Sambugaro et al., 2004).

Statistical analysis

The anatomical leaf characterization data were subjected to multivariate statistical tests of Cluster Analysis and Principal Component Analysis (PCA) (Sneath & Sokal, 1973) to verify the discriminatory capacity of the quantitative anatomical variables obtained by the measurements of the different leaf tissues, and the means compared by the Tukey test at 5% probability, using R Studio software.

Results

Damage by Glycaspis brimblecombei

The infestation of G. brimblecombei was constant with low plant mortality. Eucalyptus camaldulensis was more infested than E. urophylla and E. grandis and all G. brimblecombei nymphs died in the first instars on E. saligna without development on plants of this species. Sooty mold developed on G. brimblecombei lerps. The occurrence of leaf spot from Teratosphaeria epicoccoides was observed on E. camaldulensis, E. grandis and E. urophylla and with greater damage to E. saligna.

Anatomical leaf characterization

The percentage of upper and lower epidermis in the region of the central vein, percentage of collenchyma, thickness of the upper and lower epidermis in the internervure region and thickness of the spongy parenchyma was similar between the Eucalyptus species (Table 1). The percentage of chlorenchyma was lowest and that of phloem, xylem and the mean gland area in the central vein region was highest in E. grandis leaves than in the other Eucalyptus species (Table 1).

Table 1 Values of the 18 quantitative anatomical variables for Eucalyptus camaldulensis, Eucalyptus grandis, Eucalyptus urophylla and Eucalyptus saligna leaves infested by Glycaspis brimblecombei (Hemiptera: Aphalaridae) in a greenhouse.

Variable	E. camaldulensis	E. grandis	E. urophylla	E. saligna	
Upper epidermis (%)	2.83 ± 0.67a	2.76 ± 1.00a	3.80 ± 0.57a	3.39 ± 0.75a	
Lower epidermis (%)	2.42 ± 0.36a	3.00 ± 0.52a	4.11 ± 0.69a	3.90 ± 0.71a	
Collenchyma (%)	33.46 ± 10.69a	29.44 ± 4.17a	31.09 ± 4.30a	35.70 ± 6.42a	
Phloem (%)	13.90 ± 3.86a	24.74 ± 3.52b	14.97 ± 6.39a	17.41 ± 4.34a	
Xylem (%)	16.40 ± 0.52a	19.88 ± 3.61b	12.50 ± 3.78a	10.22 ± 2.46a	
Chlorophyll parenchyma (%)	30.12 ± 4.41a	16.03 ± 3.43b	31.36 ± 5.22a	28.11 ± 3.27a	
Glands (%)	0.87 ± 0.63a	4.15 ± 1.39c	2.17 ± 2.59b	1.26 ± 1.05b	
Total cross-sectional area (mm2)	0.61 ± 0.05a	0.57 ± 0.03a	0.31 ± 0.02a	0.46 ± 0.03a	
Total of the upper epidermis (μm)	15.94 ± 3.42a	18.44 ± 3.39b	16.56 ± 2.43b	17.19 ± 2.82b	
Total of the lower epidermis (μm)	15.31 ± 3.55a	12.19 ± 2.80a	15.31 ± 2.95a	13.75 ± 3.56a	
Upper palisade parenchyma	97.19 ± 12.16a	70.94 ± 12.65b	70.00 ± 11.02b	58.12 ± 3.37b	
Lower palisade parenchyma	78.44 ± 12.76	0.00 ± 0.00	0.00 ± 0.00	0.00 ± 0.00	
Total of spongy parenchyma (μm)	103.75 ± 26.28a	121.25 ± 14.39a	102.81 ± 10.47a	117.19 ± 14.46a	
Mesophyll thickness	279.37 ± 77.14a	192.19 ± 59.23b	172.81 ± 32.00b	175.31 ± 48.85b	
Leaf thickness (μm)	310.62 ± 52.93a	222.81 ± 50.54b	204.37 ± 23.41b	206.25 ± 58.97b	
Mean area of a gland	7.65 ± 2.60a	11.68 ± 2.19a	6.52 ± 0.73a	7.39 ± 0.92a	
Number of stomata of the upper surfaces	231.73 ± 20.57a	1.37 ± 0.06b	0.00 ± 0.00	0.00 ± 0.00	
Number of stomata of the lower surfaces	256.68 ± 23.89a	500.39 ± 35.71b	527.55 ± 21.01b	557.06 ± 29.43b	
Note:

Averages followed by the same lowercase letter per line do not differ by Tukey’s test (p ≤ 0.05).

The cluster analysis, based on the discriminatory capacity of the quantitative anatomical variables, that is, comparing the elements according to the presence or absence of certain characteristics separated the Eucalyptus species into two groups (Fig. 1) based on the low level of 0.32 on the similarity distance scale: group 1–E. saligna, E. urophylla and E. grandis; group 2–E. camaldulensis, E. saligna and E. urophylla.

Figure 1 Dendrogram of the cluster analysis of the 18 quantitative anatomical variables of the leaf of four species of Eucalyptus infested by Glycaspis brimblecombei (Hemiptera: Aphalaridae), using the Average Euclidean Distance. G1: group 1; G2: group 2. Ec: Eucalyptus camaldulensis; Es: Eucalyptus saligna; Eg: Eucalyptus grandis and Eu: Eucalyptus urophylla.

The graphic dispersion of the four Eucalyptus species showed E. saligna, E. urophylla and E. grandis forming group 1 and E. camaldulensis group 2 for the principal component analysis with contrast between these species (Y1 and Y2) (Fig. 2). The graphic dispersion of the PCA and the dendrogram of the cluster analysis, grouped the four Eucalyptus species into two main groups, based on the 18 quantitative anatomical characteristics of the Eucalyptus leaves (Fig. 2).

Figure 2 Graphic dispersion of the four species of Eucalyptus, using the first two principal components (Y1 and Y2), for the set of 18 quantitative anatomical variables of the leaves infested by Glycaspis brimblecombei (Hemiptera: Aphalaridae). G1: group 1; G2: group 2. EC: Eucalyptus camaldulensis; ES: Eucalyptus saligna; EG: Eucalyptus grandis and EU: Eucalyptus urophylla.

The correlation coefficients among the 18 quantitative anatomical variables of the Eucalyptus leaves and the first two principal components (Y1 and Y2) were found to be thickness variables of the lower palisade parenchyma, mesophyll, leaf, upper palisade parenchyma, upper epidermis, as well as the number of stomata of the upper and lower surfaces. These were the main variables that served to discriminate the four Eucalyptus species, based on the first principal component (Y1) (Table 2). The discriminatory power of the absolute value of Y1 for these variables, was high. The information retained for the second principal component (Y2) was low (26.43%), which meant that analysis of this component was unsatisfactory. The combined analysis of the first principal component (Table 2) and the graphic dispersion (Fig. 2) showed that the number of stomata on the lower side, percentage of lower epidermis, thickness of the upper epidermis, and percentage of gland in the central vein of the group 2 species (E. camaldulensis) were lower than those of the group 1 species (E. saligna, E. grandis and E. urophylla) (Table 1).

Table 2 Correlations between the 18 quantitative anatomical variables retained and accumulated in Y1 and Y2 for the leaf of Eucalyptus camaldulensis, Eucalyptus grandis, Eucalyptus urophylla and Eucalyptus saligna and the first two main components (Y1 and Y2).

Original variables	Y1	Y2	Original variables	Y1	Y2	
TLPP	0.9987	0.0492	%Ph	−0.5640	0.8062	
NUS	0.9984	0.0548	CS	0.5582	0.7198	
TL	0.9772	0.2124	TSP	−0.5534	0.5648	
NLS	−0.9762	−0.1881	TLE	0.5527	−0.7287	
TM	0.9758	0.2186	%UE	−0.4584	−0.8365	
TUPP	0.9204	0.2303	%Chl	0.3943	−0.9055	
%LE	−0.7627	−0.6427	%Col	0.2953	−0.6390	
TUE	−0.7122	0.6655	MGA	−0.2374	0.9642	
%Gl	−0.6077	0.7445	%Xy	0.2077	0.9431	
%Retained	70.17	26.43	%Accumulated	70.17	96.6	
Note:

TLPP, lower palisade parenchyma thickness; NUS, number of stomata/mm2 of upper face; TL, leaf thickness (μm); NLS, number of stomata/mm2 of the lower face in the internervural region; TM, mesophyll thickness; TUPP, upper palisade parenchyma thickness; %LE, Percentage of lower epidermis; TUE, thickness of the upper epidermis; %Gl, gland; %Ph, phloem; CS, total cross-sectional area (mm2) in the central rib region; TSP, spongy parenchyma thickness (μm); TLE, lower palisade parenchyma thickness; %UE, percentage of upper epidermis; %Chl, chlorenchyma; %Col, collenchyma; MGA, mean gland area; %Xy, xylem.

The values of the thickness characteristics of the upper and lower palisade parenchyma, mesophyll and leaf and the number of stomata on the upper surface of E. camaldulensis were higher than those for other species. The E. camaldulensis leaf profile was classified (Fig. 3). Signs of stylet insertion by G. brimblecombei nymphs were found in E. camaldulensis leaf sections, passing through the collenchyma, near the central leaf vein and the palisade parenchyma (Fig. 4).

Figure 3 Central vein region of Eucalyptus grandis (A) and Eucalyptus camaldulensis (B) and internervural of Eucalyptus grandis (C) and Eucalyptus camaldulensis (D).

Bar = 100 μm. Xy = Xylem; Ph = Phloem; Col = Collenchyma; PP = Palisade parenchyma; SP = Spongy Parenchyma; Ep = Epidermis; Gl = Oil gland; SR = Secondary Rib. *Eucalyptus grandis belongs to group 1 (less susceptible); **Eucalyptus camaldulensis belongs to group 2 (susceptible).

Figure 4 Central vein (A) and internervural (B) region of Eucalyptus camaldulensis damaged by Glycaspis brimblecombei (Hemiptera: Aphalaridae); Bar = 100 μm

Arrow: points where the insect’s stylet passes. Caption: Xy = Xylem; Ph = Phloem; Col = Collenchyma; PP = Palisade parenchyma; SP = Spongy Parenchyma; St = Stomata.

Discussion

Glycaspis brimblecombei damages young plants, from 6 months to mature ones, up to cutting age, causing serious damage throughout its cycle (Saliba et al., 2019). The damage in younger plantations, between 6 months up to 2 years, results in greater losses when compared to more mature plantations (5 years or more) (Wardlaw et al., 2018). Glycaspis brimblecombei is a sucking insect and its nymphs produce a large amount of honeydew, causing the development of sooty mold (Reguia & Peris-Felipo, 2013). Teratosphaeria epicoccoides on Eucalyptus leaves, with greater damage to E. saligna, is generally associated with stressed plants (Andjic et al., 2019).

The more intense G. brimblecombei infestation on E. camaldulensis than on other species tested here is related to its susceptibility to this insect (Firmino-Wincker et al., 2009; Ribeiro et al., 2015). The lack of development of G. brimblecombei nymphs on E. saligna plants is due to the resistance related to epicuticular wax on the leaves, reducing the presence of eggs and nymphs and the severity of G. brimblecombei infestation (Brennan et al., 2001).

The similar percentage of epidermis in the central vein region, collenchyma and epidermis thickness in the internervural region, and thickness of spongy parenchyma for the resistant and susceptible Eucalyptus species (Brennan et al., 2001; Pereira et al., 2013; Ribeiro et al., 2015), indicates that these anatomical variables are not associated with the plant resistance or susceptibility to G. brimblecombei. The percentage of chlorenchyma, responsible for photosynthesis, is lower in E. grandis leaves than in the other Eucalyptus species. This is related to a reduction of leaf area, similar to that caused by Costalimaita ferruginea (Coleoptera: Chrysomelidae) on shoots and apical parts of Eucalyptus, which may reduce chlorenchyma, impairing plant development (Santos, Gonçalves & Silva, 2016). The higher percentage of glands on E. grandis leaves in the central vein region, and phloem and xylem in the central vein than in other species may be related to the presence and production of phenolic compounds in the epidermis (Santos et al., 2008), as a result of plant defense to insect pests, including G. brimblecombei.

Differences in the number of stomata on the upper surface, and thickness of the upper and lower palisade parenchyma on E. camaldulensis due to stomata on the adaxial surface and a double layer of palisade parenchyma on both sides of its leaves. The single layer of palisade parenchyma was found only on the adaxial surface of the other species (James & Bell, 1995).

The palisade parenchyma probably does not confer resistance on Eucalyptus spp. to G. brimblecombei, because this structure is duplicated on the adaxial and abaxial surfaces of E. camaldulensis leaves and single in the adaxial surface of E. grandis, E. saligna and E. urophylla, as well as thicker, on both sides, in E. camaldulensis than in the other species. The signs of stylet insertion by G. brimblecombei nymphs through the E. camaldulensis leaf sections indicates that they passed through the parenchyma cells rather than between them. Cell-degrading proteins such as amylase, cellulase, pectinase and pectinesterase enable stylet entry into the plant tissue (Wu et al., 2021). Stomata are absent or in low numbers in the adaxial surface of E. grandis, E. saligna and E. urophylla, whereas they are present on E. camaldulensis leaf side surfaces. The total number of stomata is similar between these species, but this may explain the similar infestation on the abaxial and adaxial surfaces of E. camaldulensis compared to E. urophylla, with greater infestation on the abaxial surfaces. Stylets of G. brimblecombei nymphs penetrated the mesophyll, crossing between the guard cells of the stomata, similar to that observed for this insect in E. globulus (Brennan & Weinbaum, 2001a, 2001b) and, for this reason, stomata on both sides of E. camaldulensis may confer greater susceptibility to G. brimblecombei.

Defense strategies of Eucalyptus trees for insects include physical barriers and constitutive and inducible chemical defenses (Patton et al., 2017). The concentration and variability of terpenes, the presence of specific compounds (Silveira et al., 2021), amounts of epicuticular wax in the leaves, and the occurrence of antibiosis, related to longer insect development stages or life cycles, and/or antixenosis resistance, related to extended developmental stages due to lower food intake of insects, are characteristics normally associated with Eucalyptus resistance to G. brimblecombei (Pereira et al., 2020).

The proportional area and number of stomata occupying the epidermis may also be important for G. brimblecombei nymph infestation and to explain E. camaldulensis susceptibility to this pest. The thinner epidermis of the adaxial surface and lower percentage of epidermal tissue on the abaxial surface of E. camaldulensis leaves are possibly related to the higher susceptibility to G. brimblecombei. This is a pioneering study evaluating anatomical foliar indicators in relation to Eucalyptus pests, and allows us to better understand pest infestation patterns, and concomitantly, the morphological characteristics that normally confer resistance, such as waxy coating, trichoids, and stomata in these plants.

Conclusions

The number of stomata in the adaxial and abaxial leaf surfaces and percentage of gland area in the central vein of the leaves are related to the resistance or susceptibility of Eucalyptus plants to G. brimblecombei.

Eucalyptus grandis, E. urophylla and E. saligna, with higher values of the leaf characteristics evaluated, may be considered resistant or moderately resistant to G. brimblecombei.

Supplemental Information

Supplemental Information 1 The raw measurements.

The evaluations of the quantitative anatomical variables of the different Eucalyptus genotypes.

Click here for additional data file.

Dr. Phillip John Villani (University of Melbourne, Australia) revised and corrected the English language used in this manuscript.

Additional Information and Declarations

Competing Interests

Author Contributions

Data Availability

The authors declare that they have no competing interests.

Fernando Henrique Moreno de Oliveira Del Piero conceived and designed the experiments, performed the experiments, analyzed the data, prepared figures and/or tables, and approved the final draft.

Carlos Frederico Wilcken conceived and designed the experiments, performed the experiments, analyzed the data, prepared figures and/or tables, and approved the final draft.

Maurício Magalhães Domingues analyzed the data, prepared figures and/or tables, authored or reviewed drafts of the paper, and approved the final draft.

Ana Laura Favoreto analyzed the data, prepared figures and/or tables, authored or reviewed drafts of the paper, and approved the final draft.

Roberto Antonio Rodella conceived and designed the experiments, performed the experiments, analyzed the data, prepared figures and/or tables, and approved the final draft.

Alexandre Igor Azevedo Pereira analyzed the data, prepared figures and/or tables, authored or reviewed drafts of the paper, and approved the final draft.

Wiane Meloni Silva analyzed the data, prepared figures and/or tables, authored or reviewed drafts of the paper, and approved the final draft.

José Eduardo Serrão analyzed the data, prepared figures and/or tables, authored or reviewed drafts of the paper, and approved the final draft.

José Cola Zanuncio analyzed the data, prepared figures and/or tables, authored or reviewed drafts of the paper, and approved the final draft.

The following information was supplied regarding data availability:

The raw measurements are available in the Supplemental Files.

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
