# Peer review of "Anatomical indicators of Eucalyptus spp. resistance to Glycaspis brimblecombei (Hemiptera: Aphalaridae)"

_PeerJ, doi:10.7717/peerj.13346_

## Round 0.1 · original submission · Major Revisions

Dear Dr. Del Piero and colleagues:

Thanks for submitting your manuscript to PeerJ. I have now received three independent reviews of your work, and as you will see, the reviewers raised some concerns about the research. Despite this, these reviewers are optimistic about your work and the potential impact it will have on research studying Eucalyptus ecology and pest management. Thus, I encourage you to revise your manuscript, accordingly, taking into account all of the concerns raised by both reviewers.

You hypothesize that it is possible to compare potted plants with adult plants; however, without some preliminary data that relate young plants to mature forest ones, the results obtained are random and preliminary at best.

The research lacks some important data related to the response in terms of fitness of G. brimblecombei reared on young plants and chemical analysis that could be useful in explaining what was seen and analyzed. The use of improper botanical terms different from those used in the reference figures makes the analysis difficult to understand.

Please ensure that your figures and tables contain all of the information that is necessary to support your findings and observations.

There are many comments by the reviewers that ask for more information on specific issues; please address these.

Please note that both Reviewer 1 and Reviewer 2 kindly provided a marked-up version of your manuscript.

I look forward to seeing your revision, and thanks again for submitting your work to PeerJ.

Good luck with your revision,

-joe

Reviewer 1 ·

Basic reporting

1. I am not completely competent to evaluate because I am not a native speaker. I have found some errors and I believe that a revision of the text is needed

Experimental design

1. yes
2. Authors hypothesize that it is possible to compare potted plants with adult plants. Without some preliminary data that relate young plants to mature forest ones, the results obtained are random and, at most, preliminary.

Validity of the findings

Without some preliminary data that relate young plants to mature forest ones, the results obtained are random and, at most, preliminary.

Annotated reviews are not available for download in order to protect the identity of reviewers who chose to remain anonymous.

Reviewer 2 ·

Basic reporting

Major flaws in the introduction and discussion sections.

Experimental design

Experiments established to check the anatomical characters are fine but more experiments should have been established to validate the resistant characterstics.

Validity of the findings

This is a combination of qualitative and quantitive research. Findings are validated according to the results.

Annotated reviews are not available for download in order to protect the identity of reviewers who chose to remain anonymous.

·

Basic reporting

The work deals with an innovative and extremely important subject, since G. brimblecombei is a limiting factor in the production of eucalyptus. The article has scientific merit in looking for species resistant to the insect. However, some points should be improved and are highlighted below:

General:
a) Abstract very well described, with clear objective, concise material and methods, and well-marked results.
b) The introduction is quite compact, it needs further elaboration, supporting the work's hypothesis.
c) The literature used is outdated, only 20% of the references (6 out of 30) are from the last 5 years. I suggest a data update.
Specific:
L. 61 - Improve the connection between the sentences, explore more the productivity of eucalyptus and its importance, then describe the insect pest.
L. 76 - I suggest further supporting the hypothesis of the work. Are there other works/papers in this area? Are there studies with other insects and eucalyptus, evaluating the resistance of this plant?
L. 76-79 - Explore this information further, it is very important to support the work.
L. 83-86 - This sentence is more consistent in the introduction, also because it approach part of the work's hypothesis.

Experimental design

General:
a) Material and methods propose a differentiated but important methodology to achieve the proposed objectives.
b) Was the Quantitative variables of leaf anatomical characteristics methodology was based on any author/manuscript? If yes, please cite. If it is self-authored, it needs to be more detailed, so that it can be repeated.
Specific:
L. 87-89 - In this paragraph more details are needed, I believe that some information is missing, as it is confusing.
L. 97 – Tall or height? Please check the most suitable term
L. 100-102 - It is not clear whether the insecticide was used on all seedlings. I suggest it be restructured, detailing this information
L. 116-117 - Please put the scientific name in italics

Validity of the findings

General:
a) The work has highlighted importance for the sector, however, this importance is not clear in the justification of the work. It is very important to improve the justification, bringing out the potential, scientific and technical merit of this work;
b) Figures must be numbered in the sequence in which they appear in the text. I suggest rearranging the number of figures.
c) The titles of the tables must be clear and self-explanatory, however, they must not contain excessive information. In this case, the descriptions of the 18 variables are already in the table, being unnecessary and redundant. But in the title of table 1 it is necessary to include the material in which all this was analyzed, it is not mentioned that it was made from eucalyptus leaves. On page 17 of the pdf document, the first sentence would connote a perfect title for table 2 and in this same line it would be for table 1
d) The beginning of the discussion is confusing because it is not clear that the information is referring to the data of the present work and, therefore, comparing to the literature. It is important to make it clearer, to make it explicit when the information was observed in the present work. I reread it several times to understand that it was referring to the data of the present work. Initially it just looked like a literature review. (L. 185-196)

Specific:
L. 138 - I suggest reviewing the subtitle, as what is described in the results does not refer to quantification itself, only observations were described.
L. 142 - Please explain this sentence better
L. 149-152 – Please explain this sentence better. I suggest rewriting.
L. 153-156 - How did you come to this conclusion, especially for group 2? I suggest explaining better
Fig. 4 - I suggest taking the legend out of the graphic. Position below or leave the text box highlighted, so as not to confuse it with the data.
L. 226-227 - Does this information refer to the present work or to Brennan's article?
L. 234-236 - What is the relationship of this information with the data obtained in the present work and with the histological analyses?
L. 249-250 - Was any analysis performed in this work to reach this conclusion? I believe this information would fit more as a discussion.

Additional comments

Please check the scientific nomenclature. The first time it is mentioned in the text, the genus must be in full, as well as at the beginning of sentences. In other quotes, it should be abbreviated. Check the lines: 91, 105, 185.

Suggestion of references that may help authors (it should be noted that the reviewer is not the author or co-author of these):

1. Bush et al. (2020) 10.2989/20702620.2020.1824556
2. Jere et al. (2020) 10.1111/aje.12686
3. Pereira et al. (2021) 10.18671/SCIFOR.V48N126.20
4. Silveira et al. (2021) 10.1007/s10973-021-11027-3

---

## Round 0.2 · Minor Revisions

Dear Dr. Del Piero and colleagues:

Thanks for revising your manuscript. One reviewer is very satisfied with your revision. Great! However, there are some concerns raised by the second reviewer. Damage by psyllids is critical for mature trees as well, so please address this. Also, language needs improvement before publication, so please take another pass with English and grammar.

Please address these issues and submit a revision ASAP.

Good luck with your revision,

-joe

Reviewer 1 ·

Basic reporting

Clear and unambiguous, professional English used throughout: I am not completely competent to evaluate because I am not a native speaker, but the Authors provided a revised text by Dr. Phillip John Villani (University of Melbourne, Australia);

Literature references are sufficient field background.

The structure of the article is professional with figures and tables. Raw data shared.

As the authors themselves assert, they are self-sufficient with pioneering results according to the hypotheses.

Experimental design

It is original research within the scope of the journal.
The submission clearly defines the research question.
The investigation has been conducted rigorously.
Methods are described with sufficient information to be reproducible by another investigator.

Validity of the findings

The data on which the conclusions are provided are in an acceptable discipline-specific repository. The data are statistically sound.
Conclusions are well stated and limited to supporting results.

Additional comments

The authors clarified well the perplexities highlighted with the first review.

Reviewer 2 ·

Basic reporting

The authors have made the changes. I am still not very comfortable with the text.
For example, although abaxial and adaxial terminologies are used in some sections but in other sections, awkward usage of 'lower side' 'lower plant parts' occurs (lines 238-239).

I believe still work on language is needed.

Under 'discussion'
"Glycaspis brimblecombei is a sucking insect and its damage produces a large amount of honeydew, causing the development of sooty mold (Reguia & Peris-Felipo, 2013), increasing the humidity inside the lerps, which can reduce the nymph development and survival."
this does not make sense. Firstly damage does not produce honeydew. Honeydew is secreted by G. brimblecombei. Secondly, what is the connection between increased humidity, plant resistance, and this study?
References are not cited correctly, Brennan 2001 is not cited correctly in the text (a and b are missing).
Antibiosis and antixenosis is still not discussed enough.

Experimental design

NA

Validity of the findings

NA

---

## Round 0.3 · accepted · Accept

Dear Dr. Del Piero and colleagues:

Thanks for revising your manuscript based on the concerns raised by the reviewer. I now believe that your manuscript is suitable for publication. Congratulations! I look forward to seeing this work in print, and I anticipate it being an important resource for groups studying Eucalyptus ecology and pest management. Thanks again for choosing PeerJ to publish such important work.

Best,

-joe